# Thermodynamics and Physicochemical Properties of Immobilized Maleic Anhydride-Modified Xylanase and Its Application in the Extraction of Oligosaccharides from Wheat Bran

**DOI:** 10.3390/foods12122424

**Published:** 2023-06-20

**Authors:** Yang Zhao, Xinrui Li, Shuo Guo, Jingwen Xu, Yan Cui, Mingzhu Zheng, Jingsheng Liu

**Affiliations:** 1College of Food Science and Engineering, Jilin Agricultural University, Changchun 130118, China; yangyang102607@163.com (Y.Z.); lixinrui538455@163.com (X.L.); shuoguov@163.com (S.G.); xvjingwen959@163.com (J.X.); cyan980220@163.com (Y.C.); 2National Engineering Research Center for Wheat and Corn Deep Processing, Changchun 130118, China

**Keywords:** immobilized enzymes, kinetics, thermodynamics, wheat bran, oligosaccharide

## Abstract

Xylanases are the preferred enzymes for the extracting of oligosaccharides from wheat bran. However, free xylanases have poor stability and are difficult to reuse, which limit their industrial application. In the present study, we covalently immobilized free maleic anhydride-modified xylanase (FMA-XY) to improve its reusability and stability. The immobilized maleic anhydride-modified xylanase (IMA-XY) exhibited better stability compared with the free enzyme. After six repeated uses, 52.24% of the activity of the immobilized enzyme remained. The wheat bran oligosaccharides extracted using IMA-XY were mainly xylopentoses, xylohexoses, and xyloheptoses, which were the β-configurational units and α-configurational units of xylose. The oligosaccharides also exhibited good antioxidant properties. The results indicated that FMA-XY can easily be recycled and can remain stable after immobilization; therefore, it has good prospects for future industrial applications.

## 1. Introduction

Most of the phytochemicals of wheat kernels are contained in the bran, where dietary fiber forms the main component and thus can be used as a raw material for the extracting of oligosaccharides (XOS). XOS, as a functional sugar with a slightly sweet taste, not only stimulates the growth of probiotic bacteria, but also remains undigested; so, it is frequently applied as a functional ingredient in food products. It can also be used as an antioxidant to reduce the damage caused by free radicals; hence, it is of great medical interest.

Xylanase (XY, EC 3.2.1.8) catalyzes the β-1,4 glycosidic bond in xylans [1] and releases oligosaccharides as by-products. Therefore, XY has been widely used in other fields, particularly in the juice, textile, animal feed, and food additives industries [2]. It has also been used to break down various types of agricultural wastes to synthesize bioethanol. Despite the significant advantages of enzymes, many industrial processes have requirements for pH, temperature stability, and tolerance. Xylanase cannot meet these industrial requirements because of its disadvantages related to unstable operation and, non-reusability, and the difficulty in recycling during production. Therefore, immobilizing XY would seem to be the best way to overcome these limitations, and the process can effectively improve the enzyme’s catalytic activity and enhance its stability [3].

In contrast to free enzymes, immobilizing the enzyme or confining it to a specific space allows its repeated use and the maintenance of its catalytic activity. It also provides several advantages: easy separation from the product, reduced product contamination, enhanced operational stability, and activity in highly concentrated substrates. However, immobilizing the enzyme in specific spaces often results in leakage, and therefore, the reusability of the immobilized enzymes decreases. Irreversible binding of the enzyme to the carrier through covalent immobilization prevents its release but also effectively prevents the enzyme molecules from being leaked. For the carrier activity, screening immobilized enzyme carriers is critical.

Sodium alginate (NaAlg), a biodegradable natural anionic polysaccharide, is mainly composed of α-L-gulonic aldehyde hydrochloride and β-D mannuronic aldehyde hydrochloride. Because of its biocompatibility, it was shown [4] to be a good material in encapsulation systems in early studies and to have a wide range of applications in the pharmaceutical and food fields. NaAlg, as an immobilizer has the advantages of low cost, wide availability, easy preparation, and good biocompatibility [5,6]. NaAlg, also has extremely effective gel properties and can interact with multivalent ions, particularly with Ca^2+^, forming a non-toxic, mechanically stable alginate calcium (ATCA). However, gel beads have disadvantages, such as large pores which can reduce the embedding rate. Glutaraldehyde (GA) is a spacer for the covalent immobilization of enzymes and serves as a reactive group for covalent binding between the carrier and the enzyme.

This study aimed to improve the reusability and stability of MA-XY by covalent immobilization and to extract wheat bran oligosaccharides. GA is used as a cross-linking agent in the present study to activate calcium alginate beads to improve the operational stability of immobilized MA-XY (IMA-XY). The structural changes in the IMA-XY and the altered catalytic properties were investigated. In addition the structure and antioxidant activity of oligosaccharides were analyzed.

## 2. Materials and Methods

### 2.1. Materials

Sodium alginate (biochemical grade, transferred to fix cells, and enzymes, etc.) and anhydrous calcium chloride were purchased from Shanghai Yuanye Biotechnology Co., Ltd. (Shanghai, China). glutaraldehyde from Shanghai Maclean Biochemical Technology Co. (Shanghai, China), and beechwood xylan and 3,5-dinitrosalicylic acid from hanghai Aladdin Biochemical Technology Co. (Shanghai, China). Maleic anhydride-modified xylanase (MA-XY) and wheat bran were made in the laboratory. All other reagents and chemicals used in this study were of analytical grade.

### 2.2. MA-XY Immobilization

To prepare the ATCA carrier, NaAlg solution (0.09 M) was slowly dropped into CaCl_2_ solution (0.3 M) with continuous stirring throughout the process. The freshly prepared ATCA beads were then hardened at 4 °C for 1 h, collected by filtration and rinsed three times with distilled water to remove CaCl_2_ from the surface of the beads. To obtain cross-linked structures, the hardened beads were immersed in GA solution (0.2 M) for 2 h at 25 °C while being stirred continuously at 120 rpm. Distilled water was used to remove any uncross-linked GA from the surface of the beads in order to avoid affecting the enzyme activity. The enzyme solution (precisely 5 mL, 1 mg/mL) was then aspirated and added to 1 g of cross-linked beads, while the enzyme was immobilized for 45 min at 25 °C while being stirred at 120 rpm. The unfixed enzyme was removed by 0.2 M phosphate buffer (pH 7.0) at the end of fixation then stored at 4 °C in the dark. Recovery of the enzyme activity was calculated as follows:(1)$$Y(\%)=\frac{X_{i}}{X_{0}}\;\;\times \;100\%$$
where: X_0_ and X_i_ are the activities of the free MA-XY (FMA-XY) and immobilized MA-XY (IMA-XY), respectively.

### 2.3. Xylanase Activity Assay

FMA-XY and IMA-XY activity was measured by the reducing sugars produced by xylan hydrolysis. One mL of FMA-XY solution or 0.2 g of IMA-XY was put into one mL beech xylan and incubated at 50 °C for 30 min. The reaction mixture was mixed with 2.5 mL of 3,5-dinitrosalicylic acid (DNS) and placed in a boiling water bath for 5 min to stop the reaction. Subsequently, distilled water was added to make up to 12.5 mL before the absorbance was measured at 540 nm.

### 2.4. Characterization of Beads

#### 2.4.1. Fourier Transform Infrared Spectroscopy (FT-IR)

The chemical structures of the sodium alginate, ATCA beads, glutaraldehyde-activated ATCA beads, and IMA-XY were determined by FT-IR (Nicolet iS20), (Thermo Fisher, Madison, WI, USA), at 4000–400 cm^−1^.

#### 2.4.2. Scanning Electron Microscope (SEM)

The microstructure of the ATCA beads, glutaraldehyde-activated ATCA beads, and IMA-XY was analyzed by SEM (PW-100-011), (Delmic, Delft, The Netherlands). The samples were adhered to an aluminum table then sprayed with gold. The microstructure was analyzed by applying a 10 KV accelerating voltage. 

### 2.5. Enzymatic Properties of IMA-XY and FMA-XY

#### 2.5.1. Organic Solvent Resistance

The organic tolerance of IMA-XY and FMA-XY was determined in 0% to 25% acetonitrile, in concentrations of 0% to 50% DMSO, and in methanol solutions at 30 °C for 30 min. The organic reagents were removed from the surface by rinsing with 0.2 M phosphate buffer (pH 7.0). The enzyme activities of IMA-XY and FMA-XY were determined without organic solvent and set as 100%.

#### 2.5.2. pH Stability

The pH stability of IMA-XY and FMA-XY was determined by incubation in buffer solutions with pH values ranging from 3.0 to 9.0. The highest enzyme activity in each group was determined by measuring the enzyme activity at 100% while calculating IMA-XY and FMA-XY.

#### 2.5.3. Thermal Stability

The thermal stability of IMA-XY and FMA-XY was determined by incubation at 60 °C for 0 to 100 min and by removing the samples at 20-min intervals. The percentage of the remaining activity of IMA-XY and FMA-XY was calculated by comparison with that at 0 min of incubation.

#### 2.5.4. Kinetic Constants

The kinetic parameters of IMA-XY and FMA-XY, the Michaelis constant (K_m_) and maximum reaction velocity (V_max_), were determined at different concentrations (6.0–30.0 mg/mL). The mixture was adjusted to pH 7.0 in a water bath at 50 °C. The dynamic analysis was based on constructing a Lineweaver-Burk plot [7].

#### 2.5.5. Thermodynamic Constants

The thermal stability of IMA-XY and FMA-XY was determined over a range of −60 °C for 0, 15, 30, 45, 60 and 75 min. The residual activity of IMA-XY and FMA-XY at each temperature was calculated by comparison with that of the sample at 0 min. The results of the study were expressed as the primary thermal inactivation rate constant (K_d_), the half-life (t_1/2_) and the D-value (the time required to maintain 10% of the initial activity when incubating the enzyme under certain temperature conditions). The primary reaction rate was.
(2)$$\frac{\mathrm{dA}}{\mathrm{dt}}{=-K}_{d}\;\times \;A$$

The deactivation rate constant, K_d_ (min^−1^), was calculated using Equation (3):Ln (A_t_) = −K_d_ + ln (A_0_)(3)
where: A_0_ is sample initial activity, and; A_t_ is enzyme activity during incubation t_min_.
(4)$$t_{1/2}=\frac{\ln\;(2)}{K_{d}}$$
(5)$$D-\mathrm{Value}=\frac{\ln\;(10)}{K_{d}}$$

The temperature increase required that the D-value by reduced by one logarithmic period (Z value), which was determined from the slope of the D value plotted against temperature (°C):(6)$$\mathrm{slope}=\frac{-1}{Z}$$

The activation energy, (E_d_), for the IMA-XY and FMA-XY (kJ/mol) was calculated using the graph of ln K_d_ versus 1/T (the reciprocal of the absolute temperature), from the slope of the Arrhenius formula:(7)$$\mathrm{slope}=\frac{-E_{d}}{R}$$

The thermodynamic parameters of IMA-XY and FMA-XY were calculated as follows: Activation enthalpy (ΔH*, kJ/mol)
(8)$$\Delta H^{*}{=E}_{d}-\mathrm{RT}$$

Gibbs free energy (ΔG*, kJ/mol)
(9)$$△G^{*}=–\mathrm{RT}\;\ln(\frac{K_{d}}{K_{B}}\;\frac{h}{T})$$
entropy change (ΔS*, J/mol k)
(10)$$△S^{*}=\frac{△H^{*}-△G^{*}}{T}$$
where: T is the absolute temperature (K); R is the gas constant (8.314 J mol^−1^ k^−1^); h (11.04 × 10^−34^ J min) is the Planck constant; and K_B_ (1.38 × 10^−23^ J K^−1^) is the Boltzmann constant. 

#### 2.5.6. Reusability Assay

To determine the reusability efficiency of IMA-XY and FMA-XY, the decolorization of the activity was changed for up to 6 cycles. The mixture was incubated at 50 °C for 30 min then the gel beads were recovered and washed three times with phosphate buffer (0.2 M pH = 7.0) to remove the reaction solution residue. 

#### 2.5.7. Storage Stability

The storage stability of IMA-XY and FMA-XY was determined by storing the samples at 4 °C for 35 d, and their activity was determined every 7 d.

### 2.6. Determination of Oligosaccharides Extracted from Wheat Bran

#### 2.6.1. Hydrolysis Procedures for Oligosaccharides

The arabinoxylan was obtained from the bran as described previously [8]. The bran arabinoxylan was mixed in a NaAC buffer solution (0.1 M, pH 5.0) at a ratio of 1:100 by adding IMA-XY at 55 °C for 6 h. The IMA-XY was removed to obtain crude XOS by centrifugation; then, the supernatant was collected and lyophilized. The purified product (XOS_75_) was precipitated with 75% anhydrous ethanol in alcohol overnight at 4 °C and, centrifuged at 12,000 rpm for 5 min, and the supernatant was lyophilized.

#### 2.6.2. High-Performance Liquid Chromatography (HPLC) Analysis

XOS_75_, fucose (Fuc), rhamnose (Rha), arabinose (Ara), galactose (Gal), glucose (Glc), mannose (Man), xylose (Xyl), fructose (fru), galacturonic acid (GalA) and glucuronic acid (GlcA) were used as monosaccharide standards in the high-performance liquid chromatography (HPLC1260, Agilent Technologies, Santa Clara, CA, USA). One mg of XOS_75_ was weighed out then dissolved in 1 mL of methanol solution (containing 1 M HCl) then reacted at 80 °C for 16 h under N_2_. One milliliter of trichloroacetic acid (2 M) was added and reacted at 120 °C for 1 h. Ethanol was added and dried in a water bath at 60 °C. 1-pheny-3-methyl-5-pyrazolone (PMP) solution and (0.5 mL) and 0.5 mL of NaOH (0.3 M) were mixed thoroughly with the sample; then, precisely 0.2 mL of the sample mixture was aspirated in an Eppendorf tube. After reacting for 30 min at 70 °C, 0.1 mL of HCl (0.3 M) and 0.1 mL of H_2_O were added to the sample tube and mixed well. After adding 1 mL of dichloromethane, the PMP solution was withdrawn and the chloromethane was removed. After filtration through a 0.22 µm filter membrane, the sample was analyzed by HPLC [9].

The chromatographic conditions were as follows: Agilent Eclipse XDB-C18 column (4.6 × 250 mm); UV detector (wavelength 245 nm); mobile phase-phosphate buffer solution (PBS) (0.1 mol/L, pH 7.0 and acetonitrile = 81:19 (*v/v*); flow rate, 1.0 mL/min; and injection volume, 10 µL.

#### 2.6.3. Matrix-Assisted Laser Desorption Ionization/Time-of-Flight Mass Spectrometry (MALDI-TOF-MS) Analysis

The XOS_75_ samples were analyzed by MALDI-TOF using an Ultraflex workstation (Daltonics, Bremenbroek, Germany) equipped with a nitrogen laser (λ = 2 nm). The mixture was prepared with acetonitrile: 0.1% trifluoroacetic acid (TFA) = 30:70, (TA30). The 2,5-dihydroxybenzoic acid (DHB) solution at a final concentration of 20 mg/mL was prepared using TA30 with DHB as the substrate. The DHB solution was mixed with XOS_75_ (TA30 preparation) solution (5 mg/mL) in equal volumes. Then drops of precisely 1 µL were aspirated onto a polished stainless steel target plate, left to dry then measured using an Autoflex mass spectrometer (Bruker, Billerica, MA, USA) in the reflection mode for positive ion acquisition.

#### 2.6.4. Nuclear Magnetic Resonance Spectroscopy (NMR) Measurement

XOS_75_ (20 mg) was fully dissolved in 0.6 mL of heavy water then analyzed using an AVNEO600 NMR system (Bruker) at 25 °C, and the ^1^H NMR and ^13^C NMR spectra were collected.

#### 2.6.5. Antioxidant Activity In Vitro

(1)DPPH^·^ Radical Scavenging Assay

DPPH solution (0.2 mmol/L pH 8.0) was prepared using anhydrous ethanol. In brief, 2 mL of the DPPH solution was mixed thoroughly with 2 mL of XOS_75_ solution (0.2–1.0 mg/mL) by aspiration. The mixture was incubated for 30 min at 25 °C in the dark. The absorbance value was measured at 517 nm. The DPPH clearance was calculated as follows:(11)$${\mathrm{DPPH}}^{\cdot }\mathrm{scavenging}\;\mathrm{rate}\%=[1-\frac{M_{1}-M_{2}}{M_{0}}]$$
where: M_1_ is the absorbance value of DPPH^·+^ XOS_75_; M_2_ is the absorbance value of XOS_75_; and M_0_ is the absorbance value of DPPH·.

(2)ABTS radical scavenging assay

ABTS solution (7.0 mmol/L) was mixed with an equal volume of potassium persulfate solution (2.45 mmol/L) and left overnight in the dark. The solution was diluted to an absorbance value of 0.70 ± 0.02 at 734 nm using anhydrous ethanol. Then 0.2 mL of XOS_75_ solution was added to 2.8 mL of ABTS solution and then left for 6 min in the dark at 25 °C. The absorbance value was measured at 734 nm. The ABTS^·+^ radical scavenging rate was calculated as follows:(12)$${\mathrm{ABTS}}^{\cdot +}\mathrm{scavenging}\;\mathrm{rate}\;\%=[1-\frac{M_{1}{–M}_{2}}{M_{0}}]$$
where: M_1_ is the absorbance value of ABTS^·+^ XOS_75_; M_2_ is the absorbance value of ABTS^·+^; and M_0_ is the absorbance value of ABTS^·+^.

(3)Ferric reducing antioxidant power assay

The solution for the ferric reducing antioxidant power assay (FRAP) was prepared as follows: TPTZ (10 nmol/L in 10 mM HCl) and FeCl_3_ solution (20 nmol/L) in sodium acetate buffer (0.3 mol/L) = 1:1:10 (*v/v/v*). Thirty µL of XOS_75_ solution or Fe^2+^ standard solution was aspirated then added to 264 µL of FRAP solution and incubated for 30 min at 37 °C. Vitamin C (VC) was used as a positive control. Fe^2+^ (0–1.5 mmol/L) was used as the standard curve. The absorbance value was measured at 593 nm. The antioxidant value of XOS_75_ was expressed in mmol/L Fe^2+^.

## 3. Results

### 3.1. Effect of NaAlg and Calcium Chloride on the Immobilization of MA-XY

The enzyme preparation used in this study, MA-XY, was made in the laboratory. The carboxylate group of NaAlg can combine with the ionic bond of a positive ion to form a calcium alginate gel network with stable mechanical properties [10]. Although using NaAlg-embedded MA-XY is an economical and simple immobilization method, xylan, a polysaccharide, has poor diffusion and low processing efficiency in immobilized enzymes. In contrast, covalent immobilization leads to rapid diffusion, which reduces the inhibition products and allows better contact with the substrate. Therefore, this drawback of encapsulated MA-XY can be significantly overcome. Meanwhile, GA has an essential role as a bifunctional cross-linking agent in the immobilization process of enzymes. The aldehyde group of GA can react with the amino group of D-glucosamine, leading to the formation of Schiff bases on the gel beads to promote carrier stability, thus effectively improving the immobilization of the enzyme [11]. Thus, GA can be covalently crosslinked with enzyme protein molecules to stabilize the quaternary structure of the enzyme. It has been demonstrated that GA-activated carriers provide a favorable biocompatible surface for the immobilization of enzymes [12]. Earlier reports also showed that the cross-linking of protein molecules by GA could improve their decomposition and denaturation stability [13]. Therefore, in the present study, MA-XY was covalently immobilized with calcium alginate beads as the carrier and GA as the cross-linking agent, giving an immobilization rate of 68.41%. The enzyme was immobilized with no carrier [6], which decreased the mobility of the enzyme, but at the same time increased its stability. 

### 3.2. Characterization of Beads

#### 3.2.1. FT-IR Analysis

An analysis of the FT-IR spectra for IMA-XY and FMA-XY is presented in Figure 1. The O=H bond (3000–3600 cm^−1^) stretching vibration peak of ATCA was narrower compared with that of the NaAlg powder. This was mainly because the carboxyl and hydroxyl groups of alginate had combined with calcium ions to form the chelated structure of calcium alginate beads [14]. The intense area at 2814 cm^−1^ correlated with the stretching vibration of the -CH bond. It was found that the Ca^2+^ in CaCl_2_ replaced the Na^+^ in NaAlg, which underwent some shifts in the 1600 cm^−1^ spectral band. An increase in peak intensity was found in the 1550–1650 cm^−1^ band, representing the C-C bond formed between calcium alginate and GA, which indicated activation of the carrier by GA [15]. The spectral bands of ATCA in the range of 788–1348 cm^−1^ were sharper than those of the IMA-XY, mainly due to the interaction of IMA-XY with the ATCA beads. This result was mainly caused by the complex produced by the interaction between the beads and the enzyme, resulting in the changes observed in this spectral region. A stretching vibration of the peak was also observed at 1026 cm^−1^, indicating an aromatic carbon-hydrogen bond. The activated gel beads had a broader peak at 3419 cm^−1^ after adding MA-XY, which was mainly the NH_2_ group of the enzyme, as shown in curve (c) of Figure 1. These results indicated that the immobilization of MA-XY had been successful.

#### 3.2.2. SEM Analysis

In the present study, the surface morphology of (ATCA), GA-activated ATCA, and IMA-XY was prepared and investigated (Figure 2). The spherical ATCA gel beads under moist conditions were deformed after lyophilization. Their surfaces were rough, with many wrinkles appearing after lyophilization. This may be related to the water efflux from the ATCA structure, which also led to the weakness of the matrix structure. On the other hand, areas of expansion (ice crystals) and contraction (other components) during the freeze-drying process were present. This could have been caused by the freeze-dried beads not having time to contract, so the ice crystal area turned into a fold-like structure. As a result, the whole system changed unevenly, which led to morphological changes [16]. ATCA underwent activation by GA, with the folds disappearing to give smooth surfaces and GA particles cross-linked on the beads. The SEM images of the gel beads containing MA-XY showed a certain degree of irregularity and the presence of epitaxy. Rehman et al. [17] also reported the presence of certain particles or agglomerates on the surface of immobilized enzymes. Similar results were also reported by Dai et al. [18] and Ma et al. [19], where granular polymers were observed on lyophilized calcium alginate beads.

### 3.3. Enzymatic Properties of IMA-XY and FMA-XY

#### 3.3.1. Organic Tolerance Analysis

The tolerance of enzymes to the presence of organic solvents needs to be considered when designing industrial biocatalysts and their practical applications. The organic tolerance was studied by conducting experiments using a set concentration of acetonitrile, 0–50% methanol, and 0–50% DMSO to determine the effect on the catalytic activity of IMA-XY and FMA-XY. Figure 3 shows that the enzyme activity was inhibited even after incubation in an experiment at lower concentrations of acetonitrile. When the enzyme was subjected to the conditions of solvents of higher polarity, the catalytic activity gradually decreased as water was stripped from the enzyme surface [20]. After incubation with different concentrations of acetonitrile, the activity of the IMA-XY was higher than that of the FMA-XY. This suggested that the enzyme molecules were protected from chemical attack by the calcium alginate through a three-dimensional structure and, thus maintained activity and conformational stability. The remaining activities when treated with 50% DMSO and methanol were about 87% and 78%, respectively, for IMA-XY, but only 68.27% and 61.78%, respectively, for FMA-XY. Overall, IMA-XY exhibited a higher organic tolerance to organic solvents compared with FMA-XY because the covalent cross-linking had increased the rigidity of the enzyme, while avoiding the interaction between the enzyme and organic solvents which could lead to enzyme unfolding.

#### 3.3.2. pH Tolerance Analysis

In industrial applications, the pH value is known to be an important parameter that can modify the enzyme activity. As the enzyme was in a solution or carrier microenvironment, it interacted electrostatically with the carrier matrix [21]. Thus, enzymes can be disrupted by different charges, leading to changes in the conformation of the enzyme, which in turn changes in its optimum pH value. Alternatively, the optimal pH value can change probably because of the creation of a charged microenvironment between the enzyme and the gel beads, resulting in a change in the enzyme properties [22]. Figure 4 shows the changes in the enzymatic activities of IMA-XY and FMA-XY at the optimum pH values of 6.0 and 5.0, respectively. The optimum pH value of IMA-XY shifted 1 unit towards alkaline, mainly because the carrier had been chemically altered by the pH value, thus affecting the carrier-protein activity and interactions [23]. This may be related to the microenvironment surrounding the enzyme as well as the spatial site resistance. The interaction between the carrier and the amino acids of the enzyme also contributed to a change in the optimum pH value. In the present study, the pH stability of IMA-XY also improved, with a relative enzyme activity of more than 90% in the range of pH 3.0 to 9.0. However, the minimum value of FMA-XY was 34.26%. Bibi et al. [24] immobilized XY on agar substrates and calcium alginate beads, and observed no change in pH value. While using Eudragit S-100 as a carrier to immobilize XY, another study found that the optimum pH value increased from 5.8 to 6.3 [25]. The motifs in the protein structure can be effectively protected from interference with the structural integrity of the enzyme under different pH conditions by cross-linking between the immobilized enzyme molecules [26]. The pH stability of IMA-XY also reflected the conformational stability of the enzyme molecules in media with different pH values. The functional and structural stability of immobilized enzymes also mainly depends on the composition and structure of the immobilized carrier and the nature of the enzyme, which plays an essential role in enzyme activity.

#### 3.3.3. Thermal Resistance Analysis

As well as catalytic activity, enzymes are favored if they can maintain a high activity in organic solvents, have limited leaching, and have high-temperature tolerance. It has been shown that the matrix is stable in the 3D protein structure and that the thermal stability of the immobilized enzyme can be enhanced [27]. The thermal stability of IMA-XY and FMA-XY was investigated by incubation at 60 °C for 0–100 min in the present study. Figure 5 shows that IMA-XY retained a relatively high catalytic activity after high-temperature incubation. However, the catalytic activity of FMA-XY decreased sharply at 60 °C. In contrast, IMA-XY retained 35.27% of its original activity after 100 min of incubation. It should be noted that IMA-XY exhibited higher thermal stability than FMA-XY, indicating that MA-XY covalently cross-linked with GA had a positive effect on its thermal stability. As the increase time increased, the secondary structure of FMA-XY tended to be disrupted, leading to a gradual decrease in enzyme activity. In contrast, calcium alginate beads can protect the enzyme from inactivation under high-temperature conditions [28]. Therefore MA-XY was immobilized to protect the spatial structure and improve the structural stability of the enzyme molecule. Inter- and intramolecular covalent bonds can also be formed between the cross-linker and the enzyme molecule, which require a lot of energy to be cleaved [29]. The results of the present study agreed with the findings of other studies, in that the functional groups of enzyme molecules can form covalent bonds with the surface of the carrier, thereby increasing the rigidity of the enzyme molecule and reducing the effect of temperature on enzyme activity [30]. In the present study, thermal deactivation required more energy to overcome the covalent force after MA-XY immobilization; so, it had a better thermal resistance.

#### 3.3.4. Kinetic Constants Analysis

The kinetics of IMA-XY and FMA-XY were investigated, and the reaction rates of the enzymes were determined under different substrate concentrations. The maximum reaction rate (V_max_) and Michaelis constant (K_m_) of the enzymes were calculated by using Lineaweaver–Burk plots. Figure 6 shows that the V_max_ value of FMA-XY was 0.32 min^−1^, 1.39 times higher than that of IMA-XY. This was because IMA-XY limited the diffusion of the substrate and the flexibility required for binding [31], thus leading to a lower reaction rate for IMA-XY, which was consistent with the results of previous studies on immobilized amylase [32,33]. The slow diffusion of the reaction substrates and the interactions between them lead to the accumulation of reaction products in the enzyme microenvironment, resulting in competitive inhibition. Alternatively, the structure of the enzyme molecule was distorted due to the formation of the enzyme-substrate complex. The enzyme molecule could not interact with the substrate thus the V_max_ value was lower than that of the free enzyme [34,35]. Dhiman et al. [36]. performed kinetic studies on the immobilization of XY in SiO_2_ nanoparticles. The immobilized enzyme had lower V_max_ and higher K_m_ values and reduced substrate exposure due to the restricted active site. It was suggested that the conformational changes in the immobilized enzyme molecule and the reduced accessibility to the active site of the substrate had been caused by the carrier [37]. However, the increased K_m_ value of IMA-XY compared with that of FMA-XY indicated a reduced affinity between the IMA-XY and the substrate. A similar result was observed by Kumar et al. when immobilizing XY on inorganic hybridized nanoflowers [38]. The K_m_ values of 42.92 and 30.91 mg/mL for IMA-XY and FMA-XY, respectively, have also been reported previously for the immobilizing of XY in functionalized magnetic nanoparticles for covalent attachment [39]. The increased K_m_ of immobilized enzymes may be the result of mass transfer limitations as well as spatial effects [40], where the immobilized carriers have some spatial effect on the active site of the enzyme and therefore have a slightly lower affinity than the enzyme–substrate complex.

#### 3.3.5. Mechanical Analysis of Thermal Inactivity

The thermal stability of an enzyme can be expressed as the ability to retard the unfolding process caused by high temperatures in the absence of an enzymatic substrate [41]. The thermal stability of IMA-XY and FMA-XY was further investigated by incubation at 40–60 °C for different times. Figure 7A shows that immobilization had the effect of improving the thermal stability of the enzymes, with IMA-XY still retaining 60.08% of its initial activity after 45 min of incubation at 55 °C but FMA-XY only retained47.69%. Figure 7B shows that IMA-XY lost about half of its activity after continuing incubation at the same temperature for up to 75 min. Alagoz et al. [42] covalently immobilized XY on functionalized silica with a residual activity value of 77.2% after 24 h of incubation at 50 °C. The activity of FMA-XY decreased rapidly after 45 min of incubation at 60 °C to 20.05% of its initial activity. In contrast, the relative enzyme activity of IMA-XY remained at 38.58% of the initial activity when kept at 60 °C for 75 min. The results based on the thermal stability of the enzyme showed that the activity of IMA-XY was significantly higher than that of FMA-XY because of its enhanced structure. Similarly, Muley et al. [43] also observed the enhancement of the thermal stability of XY by its immobilization on magnetic nanoparticles functionalized with (3-aminopropyl) triethoxysilane (APTES), a result which is consistent with that of the present study.

Under high-temperature conditions, the molecular conformation of the enzyme is distorted, and its ability to maintain its structural integrity is disrupted, with the catalytic activity being irreversibly reduced. Thermal inactivity mechanics can be a good representation of the relationship between the function and structure of an enzyme at a specific temperature [44,45]. The present study determined the significant effect of temperature on the activity of IMA-XY within and on the surface of the substrate. The thermodynamics was mainly used to explain the role of immobilization in improving the rigidity and stability of the enzyme. The rate of thermal inactivation (K_d_) of IMA-XY and FMA-XY was measured over the temperature range of 40–60 °C; Figure 8 shows the logarithm of the remaining enzyme activity plotted against time, indicating that they both exhibited first-order kinetics and were linear. The thermodynamic study focused on the relationship between temperature and enzyme stability. The thermal stability parameters of IMA-XY and FMA-XY are presented in Table 1. The thermodynamic study showed that the half-life (t_1/2_) decreased with increasing temperature, while the K_d_ value also increased significantly. It was found that the smaller the K_d_ value, the higher its thermal stability [46], which implied that the enzyme catalytic activity gradually decreased under high-temperature conditions. These results showed that the K_d_ values of IMA-XY were lower than those of FMA-XY at any temperature. The determination of the half-life showed that the t_1/2_ value of IMA-XY was significantly higher than that of FMA-XY, increasing from 170.90 to 255.87 min at 45 °C and from 38.72 to 59.18 min at 60 °C. The tertiary structure of the enzyme was more rigid, mainly due to the covalent binding of the enzyme and the carrier resulting in restricted intermolecular aggregation [47]. IMA-XY also exhibited a higher D value, 239.75 min, than that of the FMA-XY, 170.54 min, at 55 °C. These results indicated that IMA-XY had a significant advantage regarding thermal stability. These results have also been reported, with some increases in t_1/2_ min and D–values after the immobilization of XY [48]. Overall, the covalent linkage between the enzyme and the immobilized carrier was formed, which reduced thermal vibrations and conformational flexibility, thus reducing the denaturation and folding of the enzyme protein [49].

#### 3.3.6. Thermodynamic Parameter Analysis

The thermal denaturation activation energies (E_d_) of IMA-XY and FMA-XY were calculated using the Arrhenius formula as 91.42 KJ/mol, respectively (Figure 9). The E_d_ of IMA-XY was significantly higher than that of FMA-XY, indicating that FMA-XY undergoes a change in molecular conformation whenever time allows and at higher temperatures, and that this change requires only a small amount of energy. A higher E_d_ value, on the other hand, indicates that more energy is required to induce enzyme denaturation. This also means that covalently immobilized enzymes encourage the formation of more compact and heat-resistant enzymes [50]. Therefore, it can be speculated that at the critical temperature for enzyme denaturation, the enzyme was stored beyond the tolerable temperature for a certain period. Then at this point, only a small amount of energy was needed to denature the enzyme completely. Zou et al. also found that the thermal stability of the enzyme was increased after immobilization [51]. The enthalpy change (ΔH*) during the thermal deactivation of IMA-XY and FMA-XY represents the energy required to denature the enzyme molecule by breaking its non-covalent bonds and hydrophobic interactions [52]. The results are shown in Table 2. ΔH* decreased with increasing temperature because the high temperature affected the enzyme activity and thermal unfolding. The ΔH* of IMA-XY decreased from 88.35 to 88.27 KJ/mol, but for FMA-XY it decreased from 81.24 to 81.08 KJ/mol. While the relatively high ΔH* value indicated that more energy was required for thermal denaturation, the ΔH* for FMA-XY was less than that for IMA-XY. This means that FMA-XY denatured at high temperatures with relatively little energy, but IMA-XY at the same temperature required more energy for denaturation. The immobilization led to a change in the conformation of the enzyme, which in turn enhanced its hardness, thus making it more stable [46].

Gibbs free energy (ΔG*) is the energy produced during the thermal deactivation of an enzyme. It also represents the change in the conformation of the enzyme molecule, or the disruption of structural bonds caused by the thermal deactivation process. For both IMA-XY and FMA-XY, the ΔG* values increased with increasing temperature, indicating that more energy was required for enzyme inactivation. In contrast, lower ΔG* values are associated with spontaneous reactions and, consequently, catalytic reactions that occur more quickly. The ΔG* values of IMA-XY were higher than those of FMA-XY throughout the measurements, especially at 60 °C, where the ΔG* values of IMA-XY increased by 0.86% compared with those of FMA-XY. The entropy of deactivation (ΔS*) is related to a certain extent to the local disorder change between the transition and ground states of the enzyme molecule. The thermal denaturation of enzymes often leads to the unfolding of the enzyme structure, accompanied by an increase in the randomness and disorder of the enzyme. Table 1 and Table 2 show that the negative value of ΔS* for IMA-XY indicates an ordered state of the enzyme, which is mainly caused by the covalent binding of the enzyme to the substrate. The Z–value is the sensitivity of the enzyme to the heat treatment time; that of FMA-XY was slightly higher, than that of IMA-XY, indicating that it was more sensitive to incubation at high temperatures. The inactivation or disorder of the enzyme under high-temperature conditions would lead to the enzyme structure being unfolded. However, immobilization limited the thermal unfolding of the enzyme, and IMA-XY had a higher thermal stability than FMA-XY.

#### 3.3.7. Reusability Analysis

The reusability of enzymes is an important indicator for improving their utilization and a key factor in making industrial applications cost-effective. The reuse stability of IMA-XY was investigated over six cycles and the measuring of the relative enzyme activity (Figure 10). The enzyme activity decreased after the second cycle, mainly because the enzyme molecules were not incompletely covalently attached to the carrier and remained on the carrier surface only because of encapsulation adsorption. Thus, when reacting with the substrate or washing, the enzyme tended to fall from the surface of the carrier and its activity decreased. The relative enzyme activity of IMA-XY also gradually decreased as the number of reuses increased. Nevertheless, the catalytic activity was 72.01% of the original after three reuse cycles and stayed at more than 50% after six cycles. This showed that IMA-XY had good reusability as well as high operational stability even though the complex temperature and acid-base environment during the catalytic reaction can lead to the disruption of the enzyme conformation. PoXyn2 xylanase was immobilized on chitosan and still retained more than 50% of its initial activity after five cycles [53]. Sandro et al. immobilized XY using an agarose quilt as a carrier, and the catalytic activity remained almost unchanged after 10 cycles [54]. Fam et al. [55] immobilized XY in gel beads and the relative enzyme activity was as low as 45% after 16 consecutive uses. In another study, immobilized cellulase using ATCA retained only 10% of its initial activity after four cycles [56], a loss of activity much higher than that shown in the results of the present study. This was attributed to the covalent linkage between the enzyme and the carrier, which reduced the enzyme leakage from the carrier. Part of the reduction in enzyme activity can be caused by the leaching of the enzyme during separation or the desorption of non-covalent enzyme molecules [57]. Therefore, the IMA-XY in the present study can be used as a catalyst for continuous production in industry leading to a reduction in production costs.

#### 3.3.8. Storage Stability Analysis

The storage stability of enzymes is one of the factors used for evaluating and selecting final products for industrial production. Therefore, for practical applications, the enzyme must have high stability during storage with no significant loss of catalytic activity [58]. The storage stability of IMA-XY and FMA-XY was determined by storing the enzymes in a phosphate buffer solution (0.2 M pH 7.0) at 4 °C and measuring their activity over 35 d. Figure 11 shows that IMA-XY retained 74.78% of its initial activity after 35 d of storage, whereas FMA-XY retained only 66.44%. This indicated that the presence of the ATCA carrier significantly improved the efficiency of the enzyme. The improved storage stability was also related to the interaction between the carrier and the enzyme, which included excellent physical contact and, neutralization of spot residues, as well as the rigidity and stability of the enzyme [59]. The enzyme storage stability improved because of its cross-linking to the ATCA carrier. During long-term storage, IMA-XY was protected from external environmental disturbances because of its strong connection to the carrier. The antimicrobial properties of alginate also increased the storage capacity of the enzyme molecular microenvironment.

#### 3.3.9. XOS_75_ Yield Analysis

The yield of dietary fiber, the main component of wheat bran, extracted by the enzymatic method in the present study was about 40.05% ± 0.32%, with a further 32.5% ± 0.15% of arabinoxylan being obtained by the alkaline extraction and alcoholic precipitation methods. Thus, wheat bran proved to be suitable as a raw material for the extraction of XOS_75_. Enzymatic digestion using IMA-XY was performed to obtain XOS, followed by purification of the crude XOS using 75% ethanol, resulting in a final yield of XOS_75_ of 35.57% ± 0.17%.

#### 3.3.10. HPLC Analysis

The arabinoxylan in wheat bran was digested using IM-XY; then, the monosaccharide composition of the digested product XOS_75_ was determined by HPLC. The chromatogram of the monosaccharide composition of XOS_75_ was shown in Figure 12 and Table 3. This shows that the monosaccharide composition of XOS_75_ is mainly xylose, glucose, and arabinose, of which xylose was the main component with a relative content of up to 57.1%, followed by glucose and arabinose with 23.2% and 17.8%, respectively. Overall, the monosaccharide composition indicated that XOS_75_ extracted by IMA-XY was configured by xylose; and it further showed that the main component of extracted XOS_75_ was xylo-oligosaccharide.

#### 3.3.11. MALDI-TOF-MS Analysis

Mass spectrometry can effectively analyze the chemical structure of oligosaccharides, and accurately determine their molecular weights. To gain insights into the structure and composition of XOS_75_, MALDI-TOF-MS was used in the present study to determine the molecular weight of XOS75 as well as to analyze its chemical composition in-depth. The MALDI-TOF-MS spectrum of XOS_75_ shown in Figure 13 and Table 4 shows peaks of XOS_75_ at m/z 701.15, 833.20, 965.25, 1097.29, 1229.33, 1361.37, and 1493.38 representing the molecular weight of XOS_75_ as well as a sodium atom. The mass difference between the peaks was 132 Da, corresponding to the glycosidic bond formed by the pentose unit minus the water molecule. It was also observed that the XOS_75_ consisted of 5–11 xylose residues with the smallest oligosaccharide being xylopentose, which appears as a Na^+^ compound at m/z 701.15. However, xylodisaccharide, xylotriose, and xylotetrasaccharide were not detected, probably because the enzymatic digestion was insufficient.

#### 3.3.12. NMR Analysis

NMR spectroscopy was used to analyze the chemical structure of XOS_75_ in-depth in this study. NMR hydrogen spectroscopy can quickly and effectively analyze the sugar structure (Figure 14), with δ 3.1 to 5.4 ppm being the characteristic signal of the xylose unit. Normally, the signals caused by the heterotrimeric protons of the xylose unit in this region can be divided into four separate groups: the intermediate xylose unit of XOS_75_, the non-reducing end, the β-reducing end, and the α-reducing end. The analysis revealed the presence of δ 5.45 ppm and δ 5.26 ppm in the head hydrogen region. The signals at δ 4.63 ppm and δ 4.54 ppm were attributed to the α- and β- reducing ends of XOS_75_, and the signal peak at 4.79 ppm was due to the chemical shift of heavy water. The signal peaks concentrated at δ 3.0–4.0 ppm, including δ 4.32, 3.97, 3.75, 3.58, 3.47, and 3.32 ppm, were attributed to the C-2 to C-6 signal peaks on the glycosidic ring.

Compared with hydrogen spectroscopy, NMR carbon spectroscopy has a relatively higher resolution, and therefore, more information can be obtained to provide a more accurate analysis of the sugar structure. Figure 15 shows that in the ^13^C-NMR spectrum of XOS_75_, no signal peak was detected in the range of δ 160–180 ppm. This indicated the absence of the glyoxalate component in XOS_75_, which is consistent with the structural analysis of the monosaccharide composition. The signals in the range of δ 55.0–105.0 ppm were mainly xylose, which showed four isomeric carbon atoms at δ 91.08 ppm, δ 95.15 ppm, δ 100.52 ppm, and δ107.0.8 ppm, showing four isomeric carbon atoms attributed to pyranose. These results from ^1^H-NMR and ^13^C-NMR indicated that XOS_75_, which was dissolved by the IMA-XY enzyme, consisted of an α-configuration xylose unit and a β-configuration xylose unit.

#### 3.3.13. XOS_75_ Antioxidant Activity Analysis

The antioxidant activity of XOS_75_ was assayed by three methods, DPPH^·^ and ABTS^·+^ free radical scavenging ability and ferric-reducing antioxidant power (FRAP).

DPPH^·^ radicals are relatively stable radicals at room temperature and are therefore, usually used to assess the scavenging activity of antioxidants. Figure 15A shows that the scavenging power of XOS_75_ on DPPH^·^ radicals was positively correlated with its concentration. The activity of DPPH^·^ radicals ranged from 34.29% to 90.35%, showing a strong antioxidant activity with an IC_50_ value of 0.33 mg/mL.

The ABTS^·+^ radical scavenging rates of XOS_75_ and VC at different concentrations are shown in Figure 15B. The results were similar to those for DPPH^·^ radical scavenging ability, which increased with increasing XOS_75_ concentration. At XOS_75_ concentrations of 0.2–1.0 mg/mL, the scavenging of ABTS^·+^ radicals by XOS ranged from 24.29% to 89.35% with an *IC_50_* value of 0.49 mg/mL.

FRAP is a colorimetric method that uses antioxidants to combine 2,4,6-tripyridyl-s-triazine complexes (Fe^3+^-TPTZ), as markers of total antioxidant performance, which can effectively respond to the reducing ability of XOS_75_. Figure 15C shows that the ferric-reducing activities at different XOS_75_ concentrations were lower than those of the VC-positive control. There was a certain dose dependence between the reducing power and the XOS_75_ concentration when the concentration of XOS_75_ was between 0.2 and 1.0 mg/mL. The iron-reducing power of XOS_75_ increased with increasing concentration, being 0.53 mmol/L at a concentration of 1 mg/mL. The results thus indicated that XOS_75_ had strong antioxidant properties.

The effect on antioxidant activity was attributed to a variety of mechanisms, including the scavenging of free radicals and peroxide catabolism [60]. The mechanism of antioxidation is related to both the type of glycosidic bond and the monosaccharide composition of the oligosaccharides. The presence of glyoxylate groups can also activate the hydrogen atom of the hetero-capital carbon and improve the hydrogen supply capacity of the antioxidant [61]. Thus, the combination of several factors affects the antioxidant activity of XOS_75_. Similarly, Zhang et al. [62] demonstrated that the antioxidant activity of XOS follows the trend of concentration variation. A study by Andrea [63] found that DP 2–6 XOS had a low scavenging capacity while DP 6–9 XOS_75_ had a high scavenging capacity.

The results based on the DPPH^·^, ABTS^·+^, and FRAP assays showed that wheat bran XOS_75_ extracted by IMA-XY exhibited significant antioxidant properties, and indicated its potential as an effective natural antioxidant.

## 4. Conclusions

In this study, FMA-XY was immobilized on ATCA gel beads based on the covalent immobilization method to improve the stability and reusability of MA-XY. IMA-XY was prepared with a high degree of stability and reusability. The kinetic and thermodynamic parameters of IMA-XY and FMA-XY were studied, with the thermal inactivation constants of IMA-XY increasing, indicating an improved thermal stability. The catalytic activity of IMA-XY also remained greater than 50% after the sixth cycle of use after immobilization. The enzyme activity of IMA-XY also remained stable after 35 d of storage. Wheat bran XOS_75_ extracted from IMA-XY exhibited excellent antioxidant properties, making it ideal for inhibiting the growth of free radicals. Thus, the preparation of XOS_75_ using IMA-XY has been developed as a potential prebiotic food supplement to help address health issues, and these results of the present study offer great promise and significance for industrial applications.

## Figures and Tables

**Figure 1 foods-12-02424-f001:**
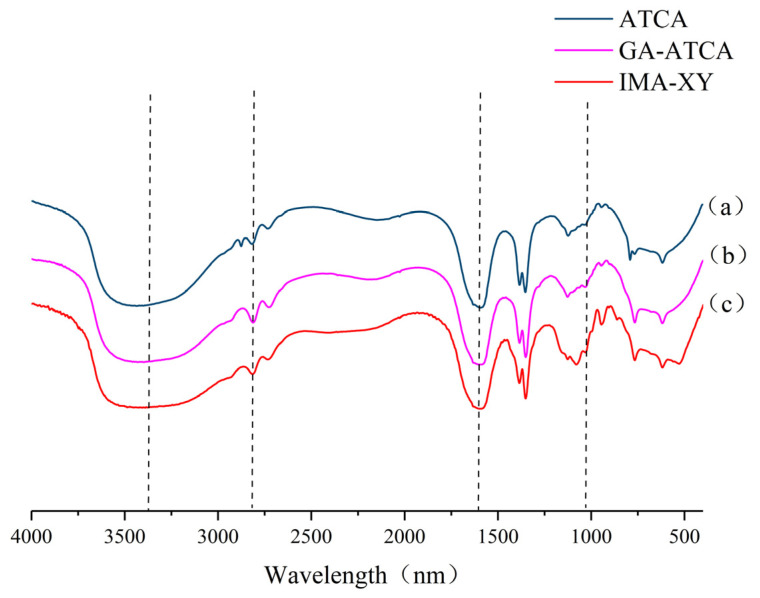
Fourier transform infrared (FT-IR) curves of IMA-XY and FMA-XY.

**Figure 2 foods-12-02424-f002:**
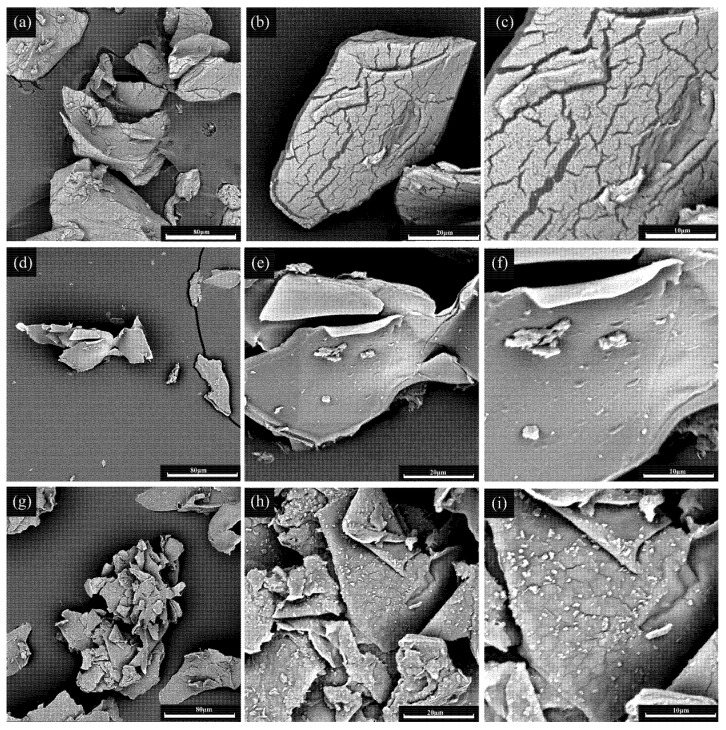
SEM images of ATCA, ATCA-GA, IMA-XY (**a**) ATCA image magnification was ×1000 (**b**) ATCA image magnification was ×4000 (**c**) ATCA image magnification was ×8000 (**d**) ATCA-GA image magnification was ×1000 (**e**) ATCA-GA image magnification was ×4000 (**f**) ATCA-GA image magnification was ×8000 (**g**) IMA-XY image magnification was ×1000 (**h**) IMA-XY image magnification was ×4000 (**i**) IMA-XY image magnification was ×8000.

**Figure 3 foods-12-02424-f003:**
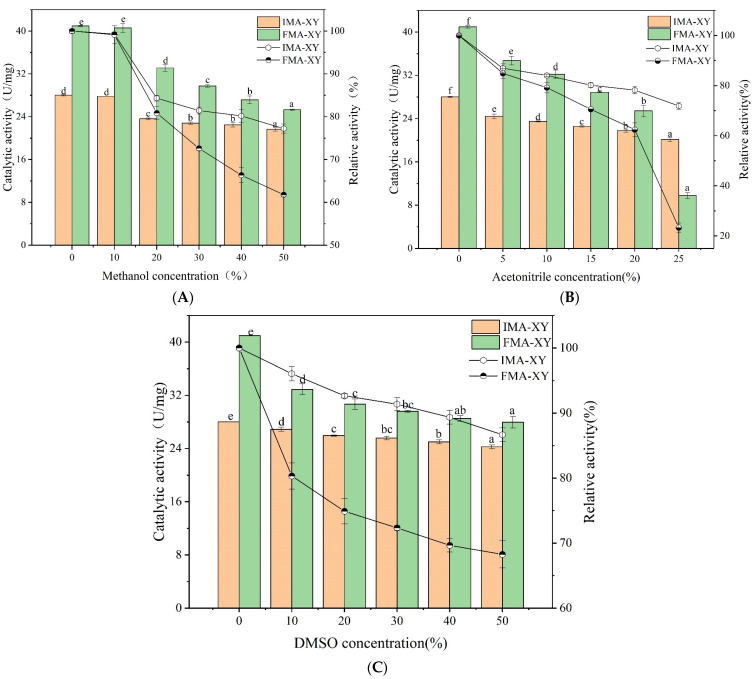
Organic tolerance profile of IMA-XY and FMA-XY; (**A**) methanol, (**B**) acetonitrile, and (**C**) DMSO. Significance analysis between groups was performed for XY and MA-XY, respectively. Different letters in the figure indicate significant differences between groups.

**Figure 4 foods-12-02424-f004:**
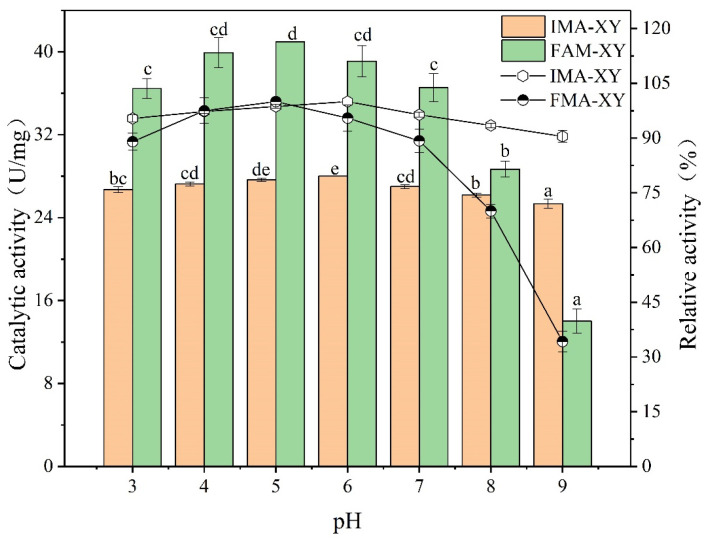
pH tolerance profile of IMA-XY and FMA-XY. Significance analysis between groups was performed for XY and MA-XY, respectively. Different letters in the figure indicate significant differences between groups.

**Figure 5 foods-12-02424-f005:**
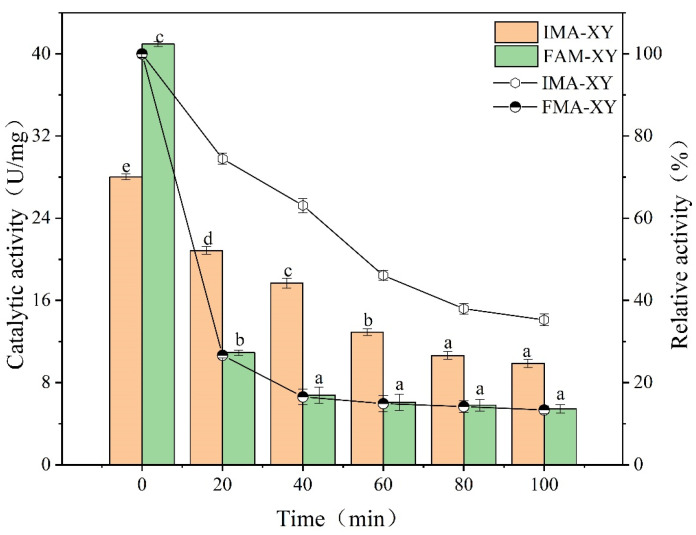
Profile of IMA-XY and FMA-XY. Significance analysis between groups was performed for XY and MA-XY, respectively. Different letters in the figure indicate significant differences between groups.

**Figure 6 foods-12-02424-f006:**
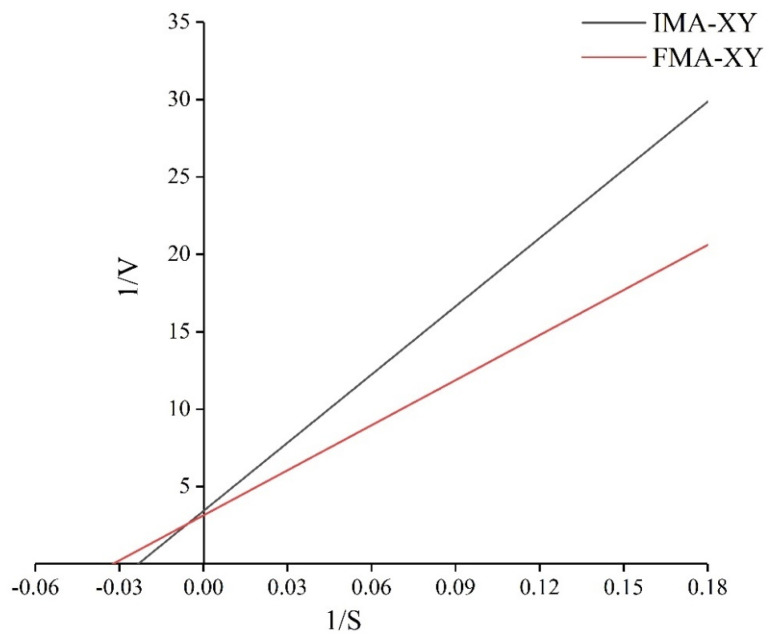
Lineweaver–Burk plots of IMA-XY and FMA-XY kinetic parameters.

**Figure 7 foods-12-02424-f007:**
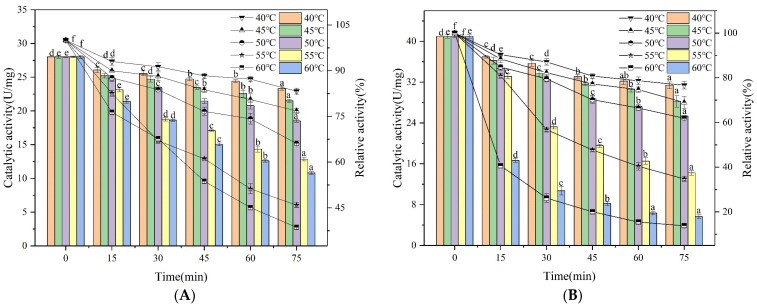
Effect of different temperature and incubation time treatments; (**A**) IMA-XY and (**B**) FMA-XY. Significance analysis between groups was performed for XY and MA-XY, respectively. Different letters in the figure indicate significant differences between groups.

**Figure 8 foods-12-02424-f008:**
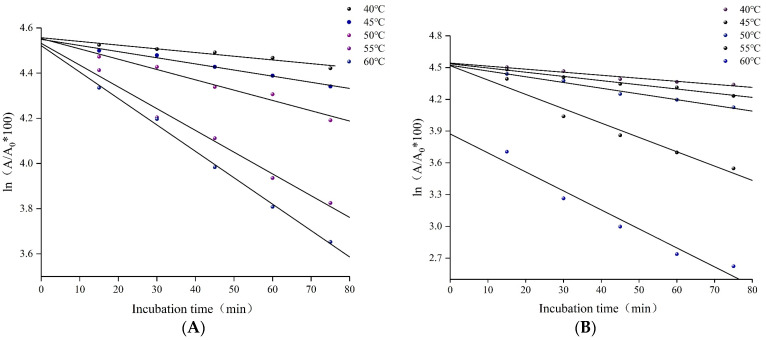
Thermal deactivation kinetics. (**A**) IMA-XY and (**B**) FMA-XY.

**Figure 9 foods-12-02424-f009:**
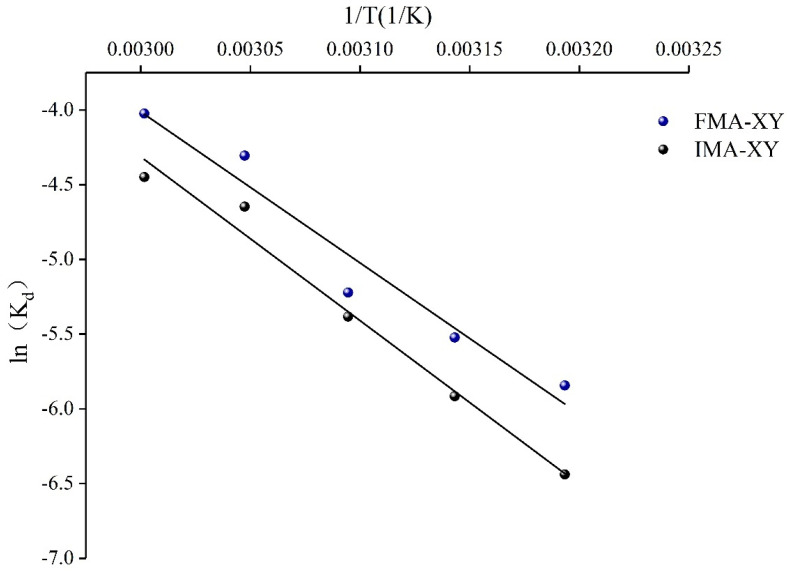
Arrhenius plots of the ln k_d_ against ratio 1/T to calculate the activation energy (E_d_) of thermal denaturation of IMA–XY and FMA–XY.

**Figure 10 foods-12-02424-f010:**
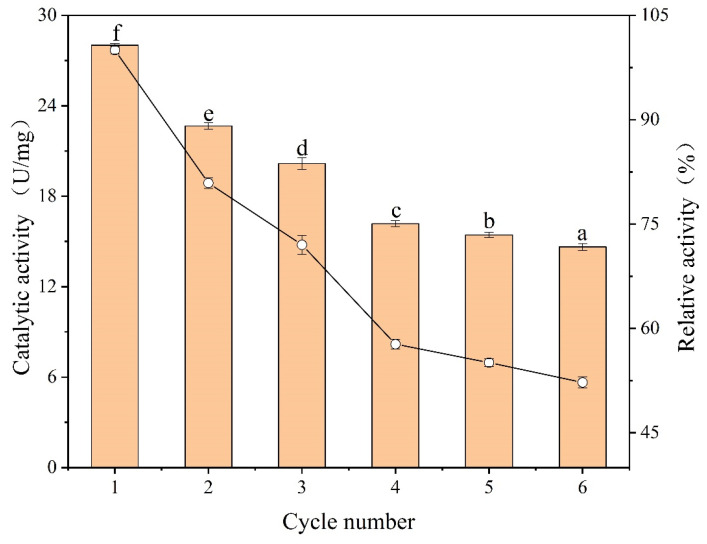
Stability of FMA-XY and FMA-XY during reusability (0.2 M phosphate buffer, pH 7.0). Note: Significance analysis between groups was performed for XY and MA-XY, respectively. Different letters in the figure indicate significant differences between groups.

**Figure 11 foods-12-02424-f011:**
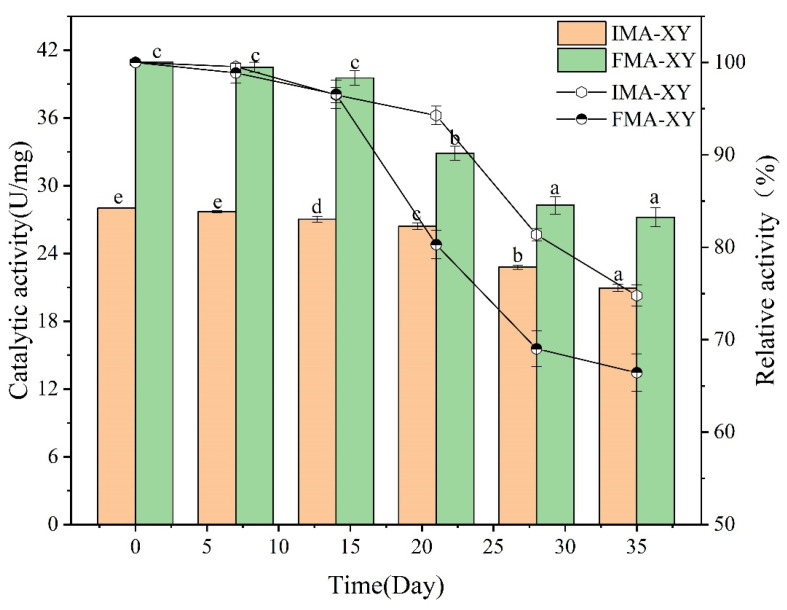
Stability of IMA-XY form during storage at 4 ± 1 °C. Note: Significance analysis between groups was performed for XY and MA-XY, respectively. Different letters in the figure indicate significant differences between groups.

**Figure 12 foods-12-02424-f012:**
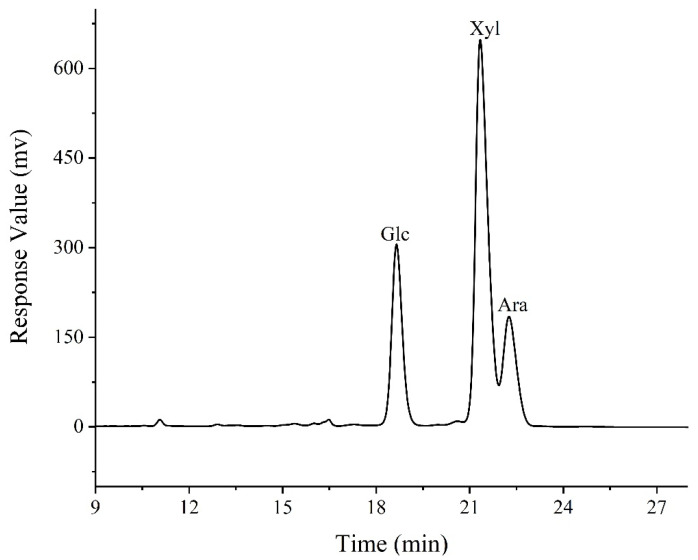
High–Performance Liquid Chromatography of XOS_75_.

**Figure 13 foods-12-02424-f013:**
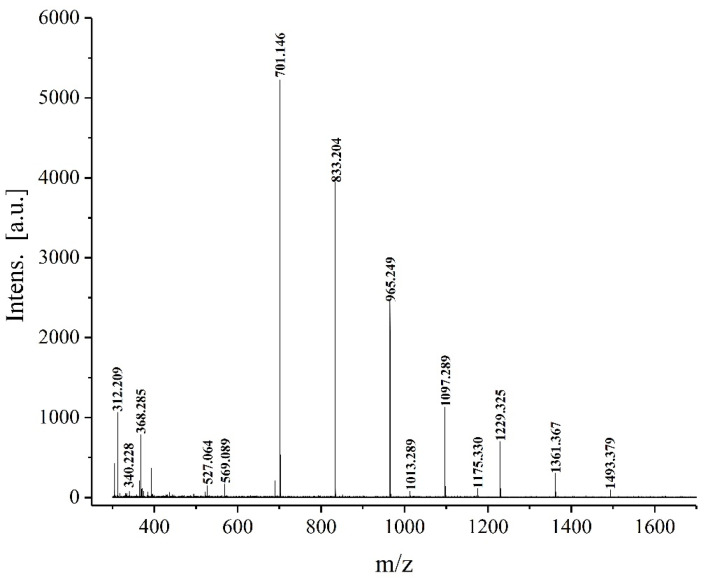
Matrix-assisted laser desorption ionization/time-of-flight mass spectrometry of XOS_75_.

**Figure 14 foods-12-02424-f014:**
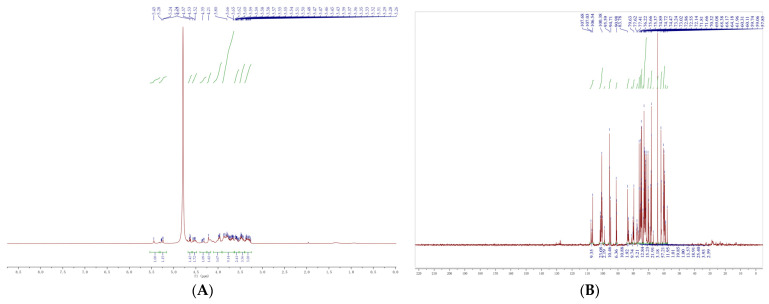
Nuclear magnetic resonance spectroscopy of XOS_75_; (**A**) ^1^H NMR and (**B**) ^13^C NMR.

**Figure 15 foods-12-02424-f015:**
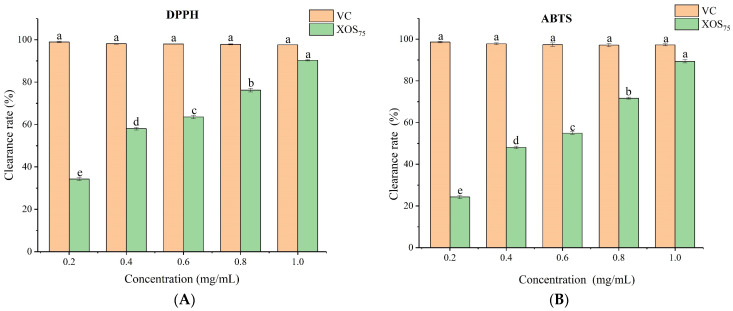
Antioxidant activity of XOS_75_; (**A**) DPPH radical scavenging activity; (**B**) ABTS radical cation scavenging activity; and (**C**) ferric-reducing antioxidant power. Different letters in the figure indicate significant differences between groups.

**Table 1 foods-12-02424-t001:** Thermal inactivation kinetics of IMA-XY and FMA-XY.

Temperature/°C	K_d_ (min^−1^)		t_1/2_ (min)		D-Value (min)	
	IMA-XY	FMA-XY	IMA-XY	FMA-XY	IMA-XY	FMA-XY
40	0.0016 ± 0.00 ^a^	0.0029 ± 0.00 ^a^	431.13 ± 2.82 ^a^	238.94 ± 0.08 ^a^	1432.17 ± 9.37 ^a^	793.73 ± 0.26 ^a^
45	0.0027 ± 0.00 ^b^	0.0027 ± 0.00 ^b^	255.87 ± 0.57 ^b^	170.90 ± 1.56 ^b^	849.98 ± 1.90 ^b^	567.72 ± 5.17 ^b^
50	0.0046 ± 0.00 ^c^	0.0046 ± 0.00 ^c^	150.42 ± 0.31 ^c^	127.23 ± 1.84 ^c^	499.69 ± 1.03 ^c^	422.65 ± 6.13 ^c^
55	0.0096 ± 0.00 ^d^	0.0135 ± 0.00 ^d^	72.17 ± 0.05 ^d^	51.33 ± 0.01 ^d^	239.75 ± 0.19 ^d^	170.54 ± 0.04 ^d^
60	0.0117 ± 0.00 ^e^	0.00179 ± 0.00 ^e^	59.18 ± 0.09 ^e^	38.72 ± 0.01 ^e^	196.60 ± 0.30 ^e^	128.63 ± 0.04 ^e^

Different letters in the same column indicate significant differences (*p* < 0.05).

**Table 2 foods-12-02424-t002:** Thermodynamic parameters of IMA-XY and FMA-XY deactivation.

Temperature/°C	ΔH* (kJ mol^−1^)		ΔG* (kJ mol^−1^)		ΔS* (J/ mol^−1^ K^−1^)	
	IMA-XY	FMA-XY	IMA-XY	FMA-XY	IMA-XY	FMA-XY
40	88.35 ± 0.30 ^a^	81.24 ± 0.15 ^a^	92.24 ± 0.02 ^b^	90.70 ± 0.00 ^a^	−12.42 ± 0.90 ^ab^	−30.21 ± 0.48 ^c^
45	88.31 ± 0.30 ^a^	81.20 ± 0.15 ^a^	92.33 ± 0.01 ^c^	91.26 ± 0.02 ^c^	−12.65 ± 0.93 ^ab^	−31.63 ± 0.40 ^b^
50	88.27 ± 0.30 ^a^	81.16 ± 0.15 ^a^	92.35 ± 0.01 ^d^	91.91 ± 0.04 ^e^	−12.66 ± 0.91 ^ab^	−33.25 ± 0.46 ^a^
55	88.22 ± 0.30 ^a^	81.12 ± 0.15 ^a^	92.63 ± 0.00 ^a^	90.85 ± 0.00 ^b^	−10.84 ± 0.91 ^b^	−29.66 ± 0.46 ^c^
60	88.27 ± 0.30 ^a^	81.08 ± 0.15 ^a^	92.27 ± 0.29 ^e^	91.46 ± 0.00 ^d^	−13.35 ± 0.91 ^a^	−31.15 ± 0.45 ^b^

Different letters in the same column indicate significant differences (*p* < 0.05).

**Table 3 foods-12-02424-t003:** High–performance liquid phase monosaccharide composition of XOS_75_.

	Monosaccharide Composition (%)					
Sample	Ara	Xyl	Glc	GlcA	Gal	Man
XOS_75_	46.2	43.1	5.6	1.7	2.6	0.8

**Table 4 foods-12-02424-t004:** Identification of signal peaks in MALDI-TOF MS.

Fraction	Ions (m/z)	Ion Structure
DP 5	701.15	[5Pent + Na]^+^
DP 6	833.20	[6Pent + Na]^+^
DP 7	965.25	[7Pent + Na]^+^
DP 8	1097.29	[8Pent + Na]^+^
DP 9	1229.33	[9Pent + Na]^+^
DP 10	1361.37	[10Pent + Na]^+^
DP 11	1493.38	[11Pent + Na]^+^

## Data Availability

Data are contained within the article. All the data generated for this study is available on request to the corresponding author.

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
