# Peer review of "Thermodynamics and Physicochemical Properties of Immobilized Maleic Anhydride-Modified Xylanase and Its Application in the Extraction of Oligosaccharides from Wheat Bran"

_foods, 2023, doi:10.3390/foods12122424_

Round 1
Reviewer 1 Report
The manuscript by Y. Zhao et al. reports the activity of immobilized enzyme. The authors evaluated the activity of immobilized enzyme from many points of view, which was good and convincing.
1. The authors mentioned in Introduction, that “Therefore, the reusability of the immobilized enzyme was reduced” (line 44). Why is it so? I think the state contradicts the sentences before/after.
2. The scale of the y-axis at least in Figure 7 and 8 should be the same for (a) and (b). It is not comfortable to compare them apparently.
3. About Figure 10, the caption should be IMA-XY, not FMA-XY, I guess? The authors mentioned that “The relative enzyme activity… as well as high operational stability”. However, 50 % of activity is not so good in terms of reusability. The sentences should be revised. Or, how about the reusability of FMA-XY? At least the comparison is required.
Author Response
Response: We appreciate the reviewer for pointing out the issue. In response, we have made appropriate adjustments to the Figure 7 and Figure 8 in the manuscript.
letter outlining the changes with respect to the reviewers' comments
We would like to show our heartfelt thanks to the reviewers for their kind and helpful comments. According with these advices, we amended the relevant part in manuscript, and someone of the questions were answered below.
To reviewer 1:
Comment 1: The authors mentioned in Introduction, that “Therefore, the reusability of the immobilized enzyme was reduced” (line 44). Why is it so? I think the state contradicts the sentences before/after.
Response: Thanks for the reviewer's careful work. we have changed sentences to “Unlike free enzymes, immobilizing the enzyme or confining it to a specific space allows its repeated use and maintenance of its catalytic activity. It also provides several advantages: easy separation from the product, reduced product contamination, enhanced operational stability, and activity in highly concentrated substrates. However, immobilizing the enzyme in specific spaces often results in leakage, and therefore, the reusability of immobilized enzymes decreases.” (Line39-44)
Comment 2: The scale of the y-axis at least in Figure 7 and 8 should be the same for (a) and (b). It is not comfortable to compare them apparently.
Response: We appreciate the reviewer for pointing out the issue. In response, we have made appropriate adjustments to the Figure 7 and Figure 8 in the manuscript.
Comment 3: About Figure 10, the caption should be IMA-XY, not FMA-XY, I guess? The authors mentioned that “The relative enzyme activity… as well as high operational stability”. However, 50 % of activity is not so good in terms of reusability. The sentences should be revised. Or, how about the reusability of FMA-XY? At least the comparison is required.
Response: We appreciate the valuable suggestions provided by the reviewer. We have recognized the oversight that led to repeated paragraphs during the revision process, resulting a misunderstanding of the statement in the title of Figure 10. To address this issue, we have modified the title of Figure 10 to read:Figure 10. Stability of IMA-XY during reusability (0.2 M phosphate buffer, pH 7.0).

Reviewer 2 Report
This paper deals with thermodynamics, physicochemical properties of immobilized maleic anhydride-modified xylanase and its application in the extraction of oligosaccharides from wheat bran. Generally, the paper is well written and is of interest to the wide range of readers. Please find below some suggestions to improve the manuscript:
Lines 94, 169 etc Please include full names for PBS, DNS, DHB etc. Check the whole manuscript for the abbreviations and add explanation when needed.
Materials and methods part are difficult to follow as many abbreviations are used and some important information are missing. This section should allow the reader to repeat your experiments. HPLC, MALDI, what instrument was used? How was calibration performed? What was used for calibration?
Add reactions where appropriate to allow the reader to follow what you had done
Author Response
letter outlining the changes with respect to the reviewers' comments
We would like to show our heartfelt thanks to the reviewers for their kind and helpful comments. According with these advices, we amended the relevant part in manuscript, and someone of the questions were answered below.
To reviewer 2:
Comment 1: Lines 94, 169 etc. Please include full names for PBS, DNS, DHB etc. Check the whole manuscript for the abbreviations and add explanation when needed.
Response: Thanks to the reviewer's suggestion, we rechecked the manuscript for all abbreviations added to the manuscript. Such as: Phosphate buffer solution (PBS); 3,5-dinitrosalicylic acid (DNS); 2,5-dihydroxybenzoic acid (DHB).
Comment 2: Materials and methods part are difficult to follow as many abbreviations are used and some important information are missing. This section should allow the reader to repeat your experiments. HPLC, MALDI, what instrument was used? How was calibration performed? What was used for calibration?
Response: We appreciate the valuable suggestions provided by the reviewer. We have clearly labeled all abbreviations with their original vocabulary and rewritten the materials and methods. HPLC is high performance liquid chromatography (HPLC1260, Agilent Technologies, Santa Clara, CA, USA), as described by Zhao [1]. MALDI-TOF is an Ultraflex workstation (Daltonics, Bremenbroek, Germany), the sample measured by Yang [2].
Comment 3: Add reactions where appropriate to allow the reader to follow what you had done
Response: We thank the reviewers for their careful work. We have re-added reactions, which are marked up in the manuscript.
References
- Zhao, B.; Wang, X.; Liu, H.; Lv, C.; Lu, J. Structural characterization and antioxidant activity of oligosaccharides from Panax ginseng C. A. Meyer. International Journal of Biological Macromolecules 2020, 150.
- Yang, Z.Y.; Wu, D.T.; Chen, C.W.; Cheong, K.L.; Deng, Y.; Chen, L.X.; Han, B.X.; Chen, N.F.; Zhao, J.; Li, S.P. Preparation of xylooligosaccharides from xylan by controlled acid hydrolysis and fast protein liquid chromatography coupled with refractive index detection. Separation & Purification Technology 2016, 151-156.

Reviewer 3 Report
The manuscript entitled “Thermodynamics, physicochemical properties of immobilized maleic anhydride-modified xylanase and its application in the extraction of oligosaccharides from wheat bran” is informative and I have some comments for the improvement of manuscript. My comments are as follows,
1. The abstract part is quite confusing. Hence, I request the authors to rewrite the abstract part in a better way.
2. It is better to highlight the claims of the work in introduction part
3. I don’t know why the authors were tried to characterize the sodium alginate which is purchased from vendor (biochemical grade). I suggest the authors to refer section 2.1
4. I haven’t fin nay wave numbers in FTIR spectral analysis data. It is too difficult to distinguish and the readers cannot differentiate it. Check Figure.1 and label it for better understanding
5. The scale bars given in the SEM images is not clear and what the authors were trying to say using the SEM image?
6. A brief discussion is required for each part.
7. The figure 11 was not clear. Hence, I suggest the authors to display it in a readable manner
8. Overall, the manuscript contains some linguistic errors and scientific clarity is missing. Hence, I recommend the authors to check and modify before submitting to any journals.
Moderate editing of english language is required.
Author Response
letter outlining the changes with respect to the reviewers' comments
We would like to show our heartfelt thanks to the reviewers for their kind and helpful comments. According with these advices, we amended the relevant part in manuscript, and someone of the questions were answered below.
To reviewer 3:
Comment 1: The abstract part is quite confusing. Hence, I request the authors to rewrite the abstract part in a better way.
Response: Thanks for the reviewer's careful work. We have rearranged the abstracts: “Xylanases are the preferred enzymes for the extracting oligosaccharides from wheat bran. However, free xylanases have poor stability and are difficult to reuse limiting their industrial application. In the present study, we covalently immobilized free maleic anhydride-modified xylanase (FMA-XY) to improve its reusability and stability. The immobilized maleic anhydride-modified xylanase (IMA-XY) exhibited better stability compared with the free enzyme. After six repeated uses, 52.24% of the activity of the immobilized enzyme remained. The wheat bran oligosaccharides extracted using IMA-XY were mainly xylopentoses, xylohexoses, and xyloheptoses, which were the β-configurational units and α-configurational units of xylose. The oligosaccharides also exhibited good antioxidant properties. The results indicated that FMA-XY can easily be recycled and remained stable after immobilization and therefore has good prospects for future industrial applications.” (line9-19)
Comment 2: It is better to highlight the claims of the work in introduction part
Response: Thanks for the reviewer's careful work. We have included claims for the study in the introduction “This study aimed to improve the reusability and stability of MA-XY by covalent immobilization and to extract wheat bran oligosaccharides. GA will be used as a cross-linking agent in the present study to activate calcium alginate beads to improve the operational stability of immobilized MA-XY (IMA-XY). The structural changes in the IMA-XY and the altered catalytic properties were investigated. And the structure and antioxidant activity of oligosaccharides were analyzed.” (Line59-64)
Comment 3: I don’t know why the authors were tried to characterize the sodium alginate which is purchased from vendor (biochemical grade). I suggest the authors to refer section 2.1
Response: We thank the experts for their careful comments, and after careful consideration, we agree that characterization is not necessary, and therefore the characterization of sodium alginate (biochemical grade) was removed from this study.
Comment 4: I haven’t fin nay wave numbers in FTIR spectral analysis data. It is too difficult to distinguish and the readers cannot differentiate it. Check Figure.1 and label it for better understanding
Response: Thanks for the reviewer's careful work. We have labeled the Fourier variation infrared spectra of Figure 1 clearly
Comment 5: The scale bars given in the SEM images is not clear and what the authors were trying to say using the SEM image?
Response: Thanks for the reviewer's careful work. We have rescaled the SEM images. The SEM images of gel beads containing MA-XY showed a certain degree of irregularity and the presence of epitaxy, again confirming that MA-XY is immobilized on the ATCA surface.
Comment 6: A brief discussion is required for each part.
Response: Thanks for the reviewer's careful work. We provide a brief discussion after the end of each section.
Comment 7: The figure 11 was not clear. Hence, I suggest the authors to display it in a readable manner.
Response: We appreciate the valuable suggestions provided by the reviewer. We have recognized the oversight that led to repeated paragraphs during the revision process, resulting a misunderstanding of the statement in the title of Figure 10. To address this issue, we have modified the title of Figure 10 to read:Figure 10. Stability of IMA-XY during reusability (0.2 M phosphate buffer, pH 7.0).
Comment 8: Overall, the manuscript contains some linguistic errors and scientific clarity is missing. Hence, I recommend the authors to check and modify before submitting to any journals.
Response: Thanks for the reviewer's careful work. We apologize for the poor language of our manuscript. We worked on the manuscript for a long time and the repeated addition and removal of sentences and sections obviously led to poor readability. We have now worked on both language and readability and have also involved native English speakers for language corrections. We really hope that the flow and language level have been substantially improved.
